# In Vitro Anthelmintic Activity of a Hydroalcoholic Extract from *Guazuma ulmifolia* Leaves against *Haemonchus contortus*

**DOI:** 10.3390/pathogens11101160

**Published:** 2022-10-07

**Authors:** Guillermo Reséndiz-González, Rosa Isabel Higuera-Piedrahita, Alejandro Lara-Bueno, Roberto González-Gardúño, Jorge Alberto Cortes-Morales, Manasés González-Cortazar, Pedro Mendoza-de Gives, Sara Guadalupe Romero-Romero, Agustín Olmedo-Juárez

**Affiliations:** 1Posgrado en Producción Animal, Departamento de Zootecnia, Universidad Autónoma Chapingo, Carr. Federal México-Texcoco Km 38.5, Texcoco CP 56230, Mexico; 2Facultad de Estudios Superiores Cuautitlán, Universidad Nacional Autónoma de México, Carr. Cuautitlán-Teoloyucan Km 2.5, Col. San Sebastián Xhala, Cuautitlán CP 54714, Mexico; 3Laboratorio de Fitoquímica y Productos Naturales, Centro de Investigación en Biodiversidad y Conservación, Universidad Autónoma del Estado de Morelos, Av. Universidad 1001, Colonia Chamilpa, Cuernavaca CP 62209, Mexico; 4Centro de Investigación Biomédica Del Sur, Instituto Mexicano del Seguro Social (IMSS), Argentina No. 1, Xochitepec CP 62790, Mexico; 5Centro Nacional de Investigación Disciplinaria en Salud Animal e Inocuidad, Instituto Nacional de Investigaciones Agrícolas, Forestales y Pecuarias (INIFAP), Carr. Fed. Cuernavaca-Cuautla No. 8534, Jiutepec CP 62550, Mexico

**Keywords:** *Guazuma*, anthelmintic activity, organic fraction, hydroxycinnamic acid, nematode, egg hatching inhibition, biological model

## Abstract

The purpose of the present study was to assess the ovicidal and larvicidal activity of a hydroalcoholic extract (HAE) and their fractions (aqueous, Aq-F and organic, EtOAc-F) from *Guazuma ulmifolia* leaves using *Haemonchus contortus* as a biological model. The egg hatching inhibition (EHI) and larval mortality against infective larvae (L3) tests were used to determine the anthelmintic effect of the treatments. The extract and fractions were tested at different concentrations against eggs and L3. Additionally, distilled water and methanol were used as negative controls and ivermectin as a positive control. The extract and fractions were subjected to HPLC analysis to identify the major compounds. The HAE displayed the highest ovicidal activity (100% EHI at 10 mg/mL). Fractionation of the HA extract allowed increasing the nematicidal effect in the EtOAc-F (100% EHI at 0.62 mg/mL and 85.35% mortality at 25 mg/mL). The phytochemical analysis of the extract and fractions revealed the presence of kaempferol, ethyl ferulate, ethyl coumarate, flavonol, luteolin, ferulic acid, luteolin rhamnoside, apigenin rutinoside, coumaric acid derivative, luteolin glucoside, and quercetin glucoside. These results suggest that *G. ulmifolia* leaves could be potential candidates for the control of *H. contortus* or other gastrointestinal parasitic nematodes.

## 1. Introduction

Gastrointestinal parasitic nematodes in sheep and goats are considered one of the main problems in extensive breading systems that severely affect the livestock industry [1,2]. There are reports on economic losses associated with gastrointestinal nematodes (GIN) in livestock worldwide [3,4]. Likewise, in a study, the influence of GIN on sheep production was evaluated through a meta-analysis and the results of this study indicate a decrease of body weight, wool production, and milk (15, 10, and 22%, respectively) with respect to uninfected animals [5]. The main strategy employed for GIN control in sheep and goats has been carried out using anthelmintic drugs. However, the excessive use of these drugs results in a high economic cost over the world and usually induces the development of resistance in GIN to most anthelmintic drugs of the same chemical group [6,7,8,9]. *Haemonchus contortus* is a parasitic nematode (Order: Strongylida) belonging to the Trichostrongylidae family. This is a highly pathogenic parasite affecting small ruminants and due to its hematophagous habits causes severe anaemia followed by emaciation and cachexia, and loss of body weight, that can lead to death [10,11,12]. For these reasons, alternative control strategies such as using plants rich in secondary metabolites are necessary. The secondary metabolites are compounds derivative of biosynthesis routes of carbon from the plant primary metabolism and have been used for different purposes i.e., food additives, antioxidants, and anthelmintics [13]. Several plants rich in secondary metabolites such as plants from Fabaceae, Asteraceae, Brassicaceae, and Malvaceae families, have been assessed as anthelmintics [14,15,16,17,18,19,20,21]. *Guazuma ulmifolia* Lam, is an arboreal species known as “Guacimo” or “Cuaulote” in Mexico. This species belongs to the Malvaceae family and has been investigated for its antioxidant, antimicrobial, antiprotozoal, and anthelmintic properties [22]. In Mexico *G. ulmifolia* leaves and fruits have been used as an extra nutritional supplement in the food of lambs [23]. *Guazuma ulmifolia* leaves contain secondary metabolites such as phenolic acids (chlorogenic acid and caffeic acid) and some flavonoids such as catechin, quercetin, and luteolin [22]. Some of these compounds have been isolated from the leaves and fruits of leguminous plants and they have shown an important anthelmintic effect against GIN including *H. contortus* [22,23,24,25]. Thus, the aim of the present study was to assess the ovicidal and larvicidal activity of a hydroalcoholic extract and two fractions (aqueous and organic) from *G. ulmifolia* leaves against *H. contortus* under in vitro conditions.

## 2. Results

### 2.1. Hydroalcoholic Extract and Fractions Yields

Macerations of 500 g of the *G. ulmifolia* leaves produced 12.48% yield of the HA-E. Then, from 100% of the integrate extract 98% and 2% yield were recorded for Aq-F and EtOAc-F, respectively.

### 2.2. Chemical Characterization of the Extract and Fractions

The HPLC chromatograms of the extract and fractions are shown in Figure 1. The analysis of UV absorption spectra of major compounds revealed the presence of kaempferol (**1**) with a retention time (rt) of 17.96 min and a UV absorption spectrum at λ_max_ of 204.0, 266.3, and 367 nm, ethyl ferulate (**2**, rt = 15.40 min; UV = 216.9, 235.7 and 324.4 nm), ethyl coumarate (**3**, rt = 14.76 min; UV = 288.6 and 312.5 nm), flavonol (**4**, rt = 13.21 min; UV = 209.9, 266.3 and 352.9 nm), luteolin (**5**, rt = 13.26; UV = 254.5, 349.4 and 417.4 nm), ferulic acid (**6**, rt = 11.48 min; UV = 219.2, 241.5 and 325.5 min), luteolin rhamnoside (**7,** rt = 10.21 min; UV = 251.0, 347.0 and 418.6 nm), apigenin rutinoside (**8**, rt = 10.03 min; UV = 267.5, 339.8 and 441.54 nm), coumaric acid derivative (**9**, rt = 9.91 min; UV = 219.2 and 319.6 nm), luteolin glucoside (**10**, rt = 9.27 min; UV = 353.3, 349.4 and 451 nm) and quercetin glucoside (**11**, rt = 9.21 mn; UV = 212.2, 255.7 and 355.3 nm). The UV absorption spectra of the extract and fractions are shown as Appendix A.

### 2.3. Egg Hatching Inhibition Test

The results of the egg hatching inhibition percentages of the HA-E and fractions as well as controls are shown in Table 1. The HA extract and the Aq fraction displayed a total egg hatching inhibitory effect at 10 mg/mL. The EtOAc fraction was the best treatment showing a total ovicidal effect with only 0.62 mg/mL of concentration.

The effective concentrations (EC_50_ and EC_90_) corresponding to the EHI test of *G. ulmifolia* hydroalcoholic extracts and their fractions are shown in Table 2. The EtOAc fraction displayed the best biological activity showing an EC_50_ of 0.08 mg/mL with a confidence interval of 0.01–0.021 mg/mL. The EC_90_ of this same treatment was 0.138 mg/mL.

### 2.4. Larval Mortality Test

*Haemonchus contortus* larval mortality percentages after exposure to HA extract, fractions and controls are shown in Table 3. The highest larvicidal effect (34.08%) was achieved with the HA highest concentration (50 mg/mL). Meanwhile, the organic fraction (EtOAc-F) showed a concentration-dependent effect ranging between 40.92–85.35 with 6.25–25 mg/mL concentration. A similar effect was observed with the Aq fraction with a 77.90% larvicidal efficacy at 50 mg/mL.

Means of lethal concentrations (LC_50_ and LC_90_) of two fractions: Aq and EtOAc are shown in Table 4. The Lethal concentrations required to cause 50% and 90% of *Haemonchus contortus* infective larvae mortality with the EtOAc fraction were: 7.69 and 30.48 mg/mL respectively.

## 3. Discussion

The results of the present study revealed evidence that the HA extract and their fractions possess anthelmintic properties, where the EtOAc fraction showed the highest activity. There are reports on the anthelmintic activity from *G. ulmifolia* extracts against different parasite groups. For instance, von Son-de Fernex et al. (2016) [19] evaluated the ovicidal activity of three extracts against *Cooperia punctata,* a parasitic nematode of cattle, and they reported an egg hatching inhibitory effect of 45.42% with an acetonic: water extract. On the other hand, the in vitro ovicidal effect of an aqueous extract of *G. ulmifolia* leaves against GIN was assessed and 48% EHI was recorded [26]. The ovicidal activity observed in our study was higher than those reported by the authors mentioned above. After analysing the effective concentrations (ovicidal activity) of the EtOAc fraction it can be observed that is 18.25 and 11.25 times more active that the Aq fraction and HA extract, respectively. The lethal concentrations (larvicidal activity) observed in this study indicate that EtOAc fraction was 3.87 times more active that the Aq fraction. Several studies of plants belonging to different plant families using hydroalcoholic extractions have displayed an important anthelmintic effect. The results found in this study with HA extract were like those studies. For example, in one study using an HA-E from grape pomace against *H. contortus* eggs an EC_50_ of 1.01 mg/mL was reported [27]. The liquid-liquid separation of the *G. ulmifolia* HA extract allowed to obtain two fractions, where the EtOAc fractions showed the highest activity against *H. contortus* eggs and infective larvae.

Likewise, there is scientific evidence that the organic fraction displays the better biological activity in other plant families like Fabaceae. In a recent study, an EtOAc fraction from another Fabaceae: *Brongniartia montalvoana* inhibited the egg hatching of small ruminant GIN by 99.1% using the lowest concentration (0.8 mg/mL) [28]. The results observed in our study revealed that the organic fraction was more effective against *H. contortus* eggs (100% ovicidal effect at 0.62 mg/mL). Other studies reported high larvicidal activity on *H. contortus* infective larvae with the same fraction, meanwhile the anthelmintic activity of an aqueous fraction has shown a low or even null effect.

For example, Zarza-Albarrán et al. (2020) [20] tested an *Acacia farnesiana* pods HA extract and their fractions (Aq-F and EtOAc-F) against *H. contortus* infective larvae and they reported only in the organic fraction a larvicidal effect. In contrast, the findings observed in the present study indicate that the Aq fraction also resulted bioactive. This could be related to the content of phenolic compounds present in both fractions. The HPLC analysis in our study showed the presence of hydroxycinnamic acid derivates (Figure 1), which have been reported with important nematocidal effect [29]. After analysing the HPLC chromatograms in both fractions, it can observe that the EtOAc fraction has a higher content of hydroxycinnamic acid derivatives than the Aq fraction. Thus, the biological activity of the organic fraction could be associated to these compounds.

On the other hand, there is a report in the literature about the antiparasitic properties of *G. ulmifolia* leaves; whose ethanolic extract at 0.05 mg/mL showed antiparasitic efficacies against parasites of importance in public health: *Trypanosoma cruzi* (63.86%), *Leishmania brasiliensis* (92.2%), and *L. infantum* (95.23%) [30]. In this same study, the authors reported the presence of gallic acid, chlorogenic acid, caffeic acid, and rosmarinic acid, as well as some flavonoids like rutin, luteolin, apigenin, and quercetin. The phytochemical analysis in our study displayed the presence of flavonoids group and hydroxycinnamic acid derivatives. There is evidence about the nematocidal activity of some phenolic acids such as gallic, chlorogenic and caffeic acids. For instance, Castillo-Mitre et al. (2017) [24] isolated caffeic acid from *Acacia cochliacantha* leaves and they observed a total ovicidal effect at 1 mg/mL on *H. contortus* eggs. Meanwhile, García-Hernández et al. (2019) [31] evaluated the ovicidal effect of gallic acid obtained both from *Caesalpinia coriaria* fruits and a commercial standard; and an important egg hatching inhibitory effect was observed (close to 100%) with only 1 mg/mL. Moreover, there is a report where chlorogenic acid (commercial standard) inhibited 100% of *H. contortus* egg hatching [32]. According to scientific evidence reported in several studies, phenolic acids obtained from plants rich in secondary metabolites have shown important antiparasitic effects and these plants/plant metabolites could be used as useful tools for controlling GIN in livestock.

Although the anthelmintic effect of these natural products is not comparable with the activities shown by anthelmintic synthetic drugs, their practical use either using the whole plant or their bioactive molecules included in the diets of the animals, could be considered a viable alternative for GIN control, as an environmentally friendly alternative.

## 4. Materials and Methods

### 4.1. Plant Material

The fresh leaves of *G. ulmifolia* (9.5 kg) were collected from Tierra Blanca municipality, Veracruz, Mexico (18°33′5.92″ N, 96°22′48.57″ W) in February 2021. A voucher specimen 11511 was authenticated by Dr Alejandro Torres-Montúfar and was deposited at the Herbarium of Facultad de Estudios Superiores Cuautitlán (FES-C) Mexico. The plant material was dried in a forced air stove (Riossa ECF125, Monterrey, NL, Mexico) at 45 °C to reach a constant weight and was ground using an industrial milled (Thomas 4 model, Philadelphia, PA, USA) to reduce the particle size to 3–5 mm [17].

### 4.2. Hydroalcoholic Extract and Fraction Obtaining

Five hundred grams of leaves were macerated using a hydroalcoholic solution (70% distilled water and 30% methanol) in a ratio weight volume of 1:10 (1 g of sample to 10 mL hydroalcoholic solution) for 48 h. After this period, the hydroalcoholic extract liquid (HA-E) was filtered using three filters (gauze, cotton, and filter paper Whatman N° 4) to obtain an extract free of material residues. Part of the liquid HA-E (100 mL) was totally concentrated by distillation under reduced pressure in a rotary evaporator (Büchi R300, 123 mbar, 90 rmp, 50 °C), and dried through lyophilization processes giving a brown powder. For another part of the liquid HA-E (4500 mL), only the methanol residues were eliminated by distillation pressure and this extract was subjected to a liquid–liquid separation using ethyl acetate (1:1 *v*/*v*). This process allowed obtaining an aqueous fraction (Aq-F) and an organic fraction (EtOAc-F), which were concentrated using the rotary evaporator under the same protocol above mentioned and dried through lyophilization processes [20].

### 4.3. Major Compound Identification by HPLC

The HA-E and their fractions (Aq-F and EtOAc-F) of *G. ulmifolia* were subjected to a chromatographic analysis by HPLC using a Waters 2695 separation module HPLC system equipped with Water 996 photodiode array detector and the Empower Pro software (Waters Corporation, Milford, MA, USA). Chemical separation was achieved in a SUPERCOSIL LC-F column (4.6 × 250 mm, i.d., 5-µm particle size; Sigma-Aldrich, Belenfonte, PA, USA). The phase consisted of a 5% trifluoroacetic acid aqueous solution as solvent A and acetonitrile as solvent B. The gradient system used was as follows: 0–1 min, 0% B; 2–3 min, 5% B; 4–20 min, 30% B; 21–23 min, 50% B; 24–25 min, 80% B; 26–27 min 100% B; 28–30 min, 0% B. The flow rate was maintained at 0.9 mL/min, and the sample injection volume was 10 μL. Absorbance was measured at 330 nm. The identification of the major compounds was established based on their UV spectra [33,34].

### 4.4. Biological Material

#### 4.4.1. *Haemonchus contortus* Eggs Recovery Procedure

The eggs of this parasite were obtained from two egg-donor lambs (23.5± 2 kg of body weight, BW), previously infected with 350 infective larvae kg/BW (INIFAP strain, Mexico). Sheep were maintained indoors in metabolic cages, and they were supplied with hay and commercial concentrate and water ad libitum. The animals were housed following the care/welfare guidelines of the Mexican Official Rule NOM-051-ZOO-1995 [35]. The collection of *H. contortus* eggs was performed according to the methodology described by Coles et al. (1992) with minor modifications [36]. Briefly, 30–50 g faeces were macerated in a mortar and pestle with clean water (400 mL) and the aqueous suspension of faecal material was filtered through four sieves (400, 140, 74 and 32 µm). Finally, the eggs recovered from the last sieve were cleaned by density gradients with 40% saccharose.

#### 4.4.2. *Haemonchus contortus* Infective Larvae Recovery Procedure

Faeces were directly obtained from the rectum of the donor’s sheep. The faecal cultures were performed in Petri dishes following the Corticelli-Lai technique for seven days [37]. After this period, the infective larvae were extracted from faecal material using the Baermann funnel technique [38]. Larvae were cleaned by density gradient with saccharose (40%) and centrifugation (3500 rpm) and were exsheathed with sodium hypochlorite at 0.187%. Finally, exsheathed third stage larvae larvae were used for the mortality assay.

### 4.5. Egg Hatch Inhibition Test (EHIT)

The ovicidal activity of the HA-E, Aq-F and EtOAc-F was carried out using 96-well microtitration plates. This experiment was performed by triplicate considering four repetitions per each assay.

Each well was considered as an experimental unit, where 100 ± 15 eggs contained in 50 µL distilled water and 50 µL of extracts or fractions were deposited in each well giving a total volume of 100 µL. The treatments were established as follows: (1) HA extract (at 1.25, 2.5, 5.00 and 10.00 mg/mL), (2) Aq fraction (at 1.25, 2.5, 5.00 and 10.00 mg/mL), (3) organic fraction (EtOAc-F, at 0.31, 0.62, 1.25, 2.50 and 5.00 mg/mL), (4) negative controls (distilled water and 2%methanol) and (5) Ivermectin (5 mg/mL) as positive control. The plates were incubated in a humid chamber at room temperature (25–28 °C) for 48 h. After this period, the egg hatching process was stopped by adding Lugol’s solution (10 µL) and the total eggs or larvae (L1 or L2) in each well were counted under optical microscopy (Motic, USA) at 4 and 10×. The egg hatching inhibition percentage (%EHI) was determined using the following formula:%EHI = [(number of eggs)/(number of larvae + number of eggs)] × 100

### 4.6. Larval Mortality Assay

The assay was performed using 96-well microtritation plates. Treatments were designed as follow: (1) HA-E (12.5, 25 and 50 mg/mL), (2) Aq-F (12.5, 25 and 50 mg/mL), (3) EtOAc-F (6.25, 12.5, and 25 mg/mL), (4) distilled water and 2% methanol as negative controls and (5) ivermectin as positive control. An aqueous suspension of 50 µL containing 100 ± 15 infective larvae was deposited into each well. Then, 50 µL aliquots of extracts and fractions as well as controls, were individually added to each well. The plates were incubated at room temperature (18–25 °C) for 48 h. After this period, the total larvae (alive or dead) of each well were counted in the microscopy. The mortality percentages were estimated based on the criteria used by Olmedo-Juárez et al. (2017) [17] using the following formula:%Mortality = [(number of dead larvae)/(number of living larvae + number of dead larvae)] × 100

### 4.7. Statistical Analysis

The data of EHI and mortality percentages were normalized using a root transformation and analysed through ANOVA based on a completely randomized design by the general linear model in SAS. Means were compared among treatments using a Tukey test at 0.05 significance. The treatments with a concentration-dependent effect were subjected to regression analysis to estimate the lethal concentrations 50 and 90 (LC_50_ and LC_90_), using the PROBIT procedure by SAS [39].

## 5. Conclusions

These results suggest that *G. ulmifolia* leaves could be potential candidates for the control of *H. contortus* or other gastrointestinal parasitic nematodes of importance for the livestock industry. Likewise, the isolation and evaluation of the metabolites contained in the bioactive fraction could be crucial for future studies focused to identify the responsible compounds of the anthelmintic activity.

## Figures and Tables

**Figure 1 pathogens-11-01160-f001:**
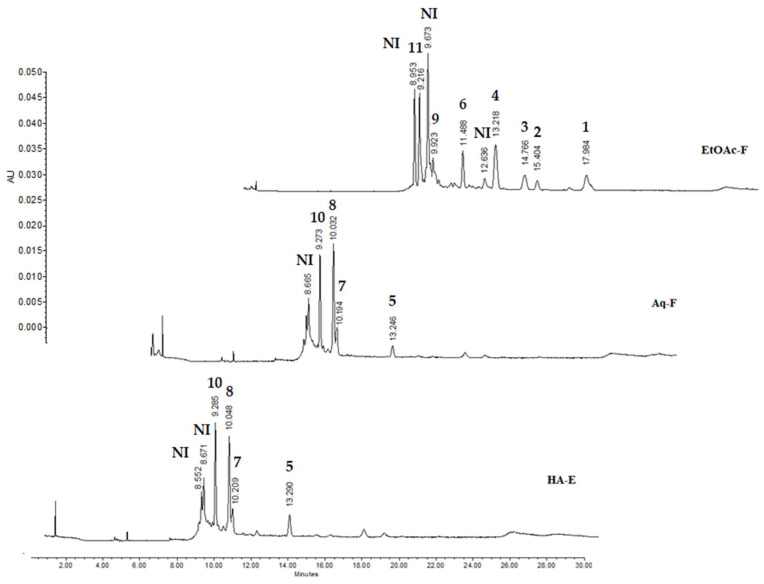
HPLC chromatograms corresponding to the hydroalcoholic extract (HA-E), aqueous fraction (Aq-F), and organic fraction (EtOAc-F), indicating the presence of kaempferol (**1**), ethyl ferulate (**2**), ethyl coumarate (**3**), flavonol (**4**), luteolin (**5**), ferulic acid (**6**), luteolin rhamnoside (**7**)**,** apigenin rutinoside (**8**), coumaric acid derivative (**9**), luteolin glucoside (**10**) and quercetin glucoside (**11**). Recorded at 350 nm.

**Table 1 pathogens-11-01160-t001:** Means of *Haemonchus contortus* eggs and larvae (L1 or L2) recovered after exposure to a *Guazuma ulmifolia* hydroalcoholic extract and fractions after 48 h incubation and egg hatching inhibition percentages.

Treatments	Mean of Recovered Nematodes		%EHI ± s.d
	Eggs	Larvae (L1 or L2)	
Distilled water	4.16	72.83	5.15 ± 5.01 ^c^
Methanol 2%	2.62	67.12	3.51 ± 3.08 ^c^
Ivermectin 5 mg/mL	81.41	0	100 ^a^
Hydroalcoholic extract (HA-E, mg/mL)			
10.0	73.5	0	100 ^a^
5.0	79.62	0.25	99.66 ± 0.61 ^a^
2.5	97.62	0.25	99.76 ± 0.65 ^a^
1.25			97.34 ± 2.11 ^a^
Aqueous fraction (Aq-F) mg/mL			
10.0	71.5	0	100 ^a^
5.0	95.5	0	100 ^a^
2.5	97.0	3.25	96.68 ± 3.17 ^a^
1.25	99.0	5.5	94.78 ± 0.75 ^a^
Organic fraction (EtOAc-F) mg/mL			
2.5	97.25	0	100 ^a^
1.25	93.0	0	100 ^a^
0.62	88.5	0	100 ^a^
0.31	65.25	20	75.49 ± 11.18 ^b^
Variation Coefficient			4.18
R^2^			0.99

^abcd^ = Means with different literal in the same column indicate statistically differences (*p* < 0.05); EHI = Egg Hatching Inhibition; L1 and L2 = First and second developing larval stages; s.d = standard deviation (n = 12).

**Table 2 pathogens-11-01160-t002:** Effective concentrations required to inhibit 50% and 90% of *Haemonchus contortus* egg hatching after 48 h exposure to a hydroalcoholic extract (HA-E) and two fractions (Aqueous Aq-F, and organic EtOAc-F) from *Guazuma ulmifolia* leaves.

Treatments	EC_50_ mg/mL	Confidence Interval (95%)		EC_90_ mg/mL	Confidence Interval (95%)	
		Lower	Upper		Lower	Upper
HA-E	0.092	0.002	0.269	0.502	0.104	0.831
Aq-F	0.146	0.028	0.300	0.923	0.544	1.204
EtOAc-F	0.008	0.001	0.021	0.138	0.086	0.187

**Table 3 pathogens-11-01160-t003:** *Haemonchus contortus* infective larvae (L3) mortality after exposure to different concentrations of *Guazuma ulmifolia* hydroalcoholic extract and their fractions expressed as percentage.

Treatments	Means of Recovered Infective Larvae		%Mortality ± s.d
	Dead	Alive	
Distilled water	0	113.5	0 ^e^
Methanol 2%	0.67	92.67	0.73 ± 0.63 ^e^
Ivermectin 5 mg/mL	51.25	0	100 ^a^
Hydroalcoholic extract (HA-E, mg/mL)			
50.0	24.00	54.25	34.08 ± 10.81 ^cd^
25.0	23.27	77.00	22.04 ± 7.23 ^cd^
12.5	22.25	83.25	21.91 ± 5.66 ^cd^
Aqueous fraction (Aq-F, mg/mL)			
50.0	79.00	23.25	77.90 ± 8.28 ^b^
25.0	28.33	84.33	25.26 ± 16.29 ^cd^
12.5	19.50	81.00	20.74 ± 9.80 ^d^
Organic fraction (EtOAc-F, mg/mL)			
25.0	100.75	17.00	85.35 ± 5.01 ^ab^
12.5	72.25	31.00	69.77 ± 2.84 ^b^
6.25	39.25	55.50	40.92 ± 9.06 ^c^
Variation coefficient			17.81
R^2^			0.96

^abcde^ Means with different literal in the same column indicate statistically differences (*p* < 0.05), s.d = standard deviation.

**Table 4 pathogens-11-01160-t004:** Lethal concentrations required to cause 50% and 90% of *Haemonchus contortus* infective larvae mortality (LC_50_ and LC_90_) after 48 h exposure to an aqueous fraction (Aq-F) and to an organic fraction (EtOAc-F) from *Guazuma ulmifolia* leaves.

Treatments	LC_50_ mg/mL	Confidence Interval (95%)		LC_90_ mg/mL	Confidence Interval (95%)	
		Lower	Upper		Lower	Upper
Aq-F	29.77	25.91	33.54	99.77	84.15	125.86
EtOAc-F	7.69	6.84	8.48	30.48	26.42	36.66

## Data Availability

Data are contained within the article.

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
