# Peer review of "In Vitro Anthelmintic Activity of a Hydroalcoholic Extract from Guazuma ulmifolia Leaves against Haemonchus contortus"

_pathogens, 2022, doi:10.3390/pathogens11101160_

Round 1
Reviewer 1 Report
The hypotheses used are good, but not sufficient. I believe you should also do tests using major compounds found in these extracts. The egg hatch inhibition test methodology is wrong. You should do more repetitions per concentration, repeat the test and do more concentrations. I don't think the larval mortality assay you used is the most appropriate methodology. I recommend that you use methodologies used and described by Hervé Hoste and Felipe Torres-Acosta.
Author Response
Reviewer
The hypotheses used are good, but not sufficient. I believe you should also do tests using major compounds found in these extracts.
Response: Thanks for your comment.
We have been working in this issue and we hope we will include this information in our next manuscript
Reviewer
The egg hatch inhibition test methodology is wrong. You should do more repetitions per concentration, repeat the test and do more concentrations
Response: Thanks for your comment. I am sorry, I think we needed to be more accurate in our description.
We would like to clarify that we performed our experiments by triplicate each and we considered four repetitions per each experiment. This give us a total of twelve true repetitions distributed into three experiments. This information has been inserted into the Materials and Methods section, page 9, lines 266-267.
We actually, followed the procedure described by Coles et al., 1992.
Reviewer
I don't think the larval mortality assay you used is the most appropriate methodology. I recommend that you use methodologies used and described by Hervé Hoste and Felipe Torres-Acosta.
Response: Thanks for your comment.
On this regard, the methodology we used is the one that we have established in our laboratory and this procedure has been accepted in a number of international journals. We think that the methodology of Drs. Hervé Hoste and Felipe Torres-Acosta is fine; although, we have got very good results with ours.
We have included some of our recent papers where we have published the same methodology.
Jasso Díaz, G., Hernández, G.T., Zamilpa, A., Becerril Pérez, C.M., Ramírez Bribiesca, J.E., Hernández Mendo, O., Sánchez Arroyo, H., González Cortazar, M., Mendoza de Gives, P. In vitro Assessment of Argemone mexicana, Taraxacum officinale, Ruta chalepensis and Tagetes filifolia against Haemonchus contortus Nematode Eggs and Infective (L3) Larvae. Microb. Pathog. 2017, 109, 162–168, https://doi.org/10.1016/j.micpath.2017.05.048.
Zarza-Albarrán, M.A.; Olmedo-Juárez, A.; Rojo-Rubio, R.; Mendoza-de Gives, P.; González-Cortazar, M.; Tapia-Maruri, D.; Mondragón-Ancelmo, J.; García-Hernández, C.; Blé-González, E.A.; Zamilpa, A. Galloyl Flavonoids from Acacia farnesiana Pods Possess Potent Anthelmintic Activity against Haemonchus contortus Eggs and Infective Larvae. J. Ethnopharmacol. 2020, 249, https://doi.org/10.1016/j.jep.2019.112402
Olmedo-Juárez, A.; Rojo-Rubio, R.; Zamilpa, A.; Mendoza de Gives, P.; Arece-García, J.; López-Arellano, M.E.; von Son- de Fernex, E. In vitro Larvicidal Effect of a Hydroalcoholic Extract from Acacia cochliacantha Leaf against Ruminant Parasitic Nematodes. Vet. Res. Commun. 2017, 41, 227–232, https://doi.org/10.1007/s11259-017-9687-8.
Reviewer 2 Report
The manuscript “In vitro Anthelmintic Activity of a Hydroalcoholic Extract from Guazuma ulmifolia Leaves Against Haemonchus contortus” contributes to the knowledge about the activity of hydroalcoholic extract and fractions of this plant. The described results are promising and provide a significant contribution that fits into scope of this scientific journal. The paper is clear, but the results can be described more extensively in the text. Some gaps have not yet been clarified:
1. The end of the introduction section should be revised considering as well other studies where Isatis tinctoria (doi: 10.3390/vetsci9030129.) and other plant (doi: 10.1016/j.rvsc.2019.01.004.) extracts were used against Ewes' Gastrointestinal Nematodes (GINs)
2. The authors should explain which compounds of the hydroalcoholic extract are present in this plant. In the results a figure 1 is mentioned in which the results obtained with HPLC analysis should be shown. However, figure 1 does not appear in the manuscript and the results are not well described in the text. I would suggest to improve the manuscript describing the composition of the aqueous and organic fractions.
3. It should be evaluated, at the end of the EHT, how many eggs were embryonic and how many were not. This is needed to understand at what stage of development the extract or fraction tested is active. This would serve to understand if it only blocks hatching or development. Also, please describe the method used to evaluate the eggs’ death (“ovicidal effect” mentioned in lines 86 – 87)?
4. How many replicates do you performed for each treatments tested?
5. How do you choose the dosage to test? Did you try the activities of plant extract concentration less than 1.25 mg/ml (HA-E), 1.25 (Aq-F), and 0.31 mg/ml (EtOAc-F)? Why?
6. Line 245: Please, delete the (1). I think it's a typo.
Author Response
Reviewer
The manuscript “In vitro Anthelmintic Activity of a Hydroalcoholic Extract from Guazuma ulmifolia Leaves Against Haemonchus contortus” contributes to the knowledge about the activity of hydroalcoholic extract and fractions of this plant. The described results are promising and provide a significant contribution that fits into scope of this scientific journal. The paper is clear, but the results can be described more extensively in the text. Some gaps have not yet been clarified:
Response: Suggestions attended
Reviewer
1. The end of the introduction section should be revised considering as well other studies where Isatis tinctoria(doi: 10.3390/vetsci9030129.) and other plant (doi: 10.1016/j.rvsc.2019.01.004.) extracts were used against Ewes' Gastrointestinal Nematodes (GINs)
Response: According to your kind recommendation, we revised some related literature and we have inserted some information into the text in the Introduction section, page 2, line 60.
The reference “doi: 10.3390/vetsci9030129” you kindly suggested was included in the new version of the manuscript.
Reviewer
2. The authors should explain which compounds of the hydroalcoholic extract are present in this plant. In the results a figure 1 is mentioned in which the results obtained with HPLC analysis should be shown. However, figure 1 does not appear in the manuscript and the results are not well described in the text. I would suggest to improve the manuscript describing the composition of the aqueous and organic fractions.
Response: We regret not to include Figure 1. Sorry!
The HPLC missing chromatograms of the extract and fractions were included in the new manuscript.
The information about the UV absorption spectra in both the extract and fractions was now included as supplementary material.
Reviewer
3. It should be evaluated, at the end of the EHT, how many eggs were embryonic and how many were not. This is needed to understand at what stage of development the extract or fraction tested is active. This would serve to understand if it only blocks hatching or development. Also, please describe the method used to evaluate the eggs’ death (“ovicidal effect” mentioned in lines 86 – 87)?
Response: We think your recommendation could have enriched our study. Unfortunately, we only performed a total count of the eggs without considering embryonic and not embryonic eggs.
The methodology employed to test the egg hatching inhibition test was according to Coles et al., 1992.
To better understand, the term “ovicidal effect” was replaced by “egg hatching inhibition percentages”.
Reviewer
4. How many replicates do you performed for each treatments tested?
Response: Each assay was tested by triplicate with four repetitions (n=4). This issue give us a total of twelve repetitions distributed into three experiments.
This information was included in the new version of the manuscript. This information has been inserted into the Materials and Methods section, page 9, lines 266-267.
Reviewer
5. How do you choose the dosage to test? Did you try the activities of plant extract concentration less than 1.25 mg/ml (HA-E), 1.25 (Aq-F), and 0.31 mg/ml (EtOAc-F)? Why?
Response: We actually have established the following procedure: First of all we normally make a screening of our extracts or fractions at different concentrations (preliminary studies). Then, when we identify one bioactive concentration. We considered this concentration as our starting point. This is the way we defined our different concentrations.
Reviewer
Line 245: Please, delete the (1). I think it's a typo.
Response: Sorry, this error was corrected
Reviewer 3 Report
Gastrointestinal nematodes are among the most prevalent disease-causing agents worldwide that infect humans, companion, and farm animals. The paper discusses the potential use of Guazuma ulmifolia as a potential anthelmintic treatment against H. contortus. To determine the anthelmintic activity of G. ulmifolia, hydroalcoholic extract with two fractions (water fraction and organic) were tested against H. contortus egg hatch and L3 mortality assays.
The paper is useful and suitable for publication after addressing minor issues.
Reviewer’s remarks to the authors:
The paper is well written in general, however some issues need to be addressed before publication.
-The abstract is very confusing and needs revision and rephrasing. For example, including doses tested in the abstract is unnecessary that may add some confusion.
-Figure 1, for HPLC graph is surprisingly missing from the draft I received.
-Table 1, column 2 and three, is quite confusing and needs revisions giving more details on how the mean of eggs and larvae (L1 or L2) recovered was calculated.
- Ivermectin is very potent at the nanomolar concentrations, however the authors used 5mg/ml of ivermectin (equal to 5.7 mM) they didn’t see any activity against egg hatch. The author needs to discuss such discrepancies of ivermectin activity.
- Table 1, the authors reported after incubation of ~100 eggs with ivermectin the mean of recovered eggs is 0 and recovered larvae is 81 which mean that the % EHI should be 0. However, the table incorrectly indicate 100% for ivermectin %EHI. Please explain.
- in the egg hatch inhibition test section in the material and methods the authors mentioned using thiabendazole at 0.1mg/ml as positive control (not ivermectin) however, no data was shown for thiabendazole.
- the paper needs another round of English revision for better reading
Author Response
Reviewer
Gastrointestinal nematodes are among the most prevalent disease-causing agents worldwide that infect humans, companion, and farm animals. The paper discusses the potential use of Guazuma ulmifolia as a potential anthelmintic treatment against H. contortus. To determine the anthelmintic activity of G. ulmifolia, hydroalcoholic extract with two fractions (water fraction and organic) were tested against H. contortus egg hatch and L3 mortality assays.
The paper is useful and suitable for publication after addressing minor issues.
Reviewer’s remarks to the authors:
The paper is well written in general, however some issues need to be addressed before publication.
-The abstract is very confusing and needs revision and rephrasing. For example, including doses tested in the abstract is unnecessary that may add some confusion.
Response: The abstract was restructured according to your kind suggestions. Thanks!
Reviewer
-Figure 1, for HPLC graph is surprisingly missing from the draft I received.
Response: I regret, I omitted this Figure. I am really sorry!
This figure with the HPLC chromatograms has been included into the new version of the manuscript.
Reviewer
-Table 1, column 2 and three, is quite confusing and needs revisions giving more details on how the mean of eggs and larvae (L1 or L2) recovered was calculated.
Response: These data correspond to the mean (n=12) of the total count either eggs or larvae (larvae 1 and larvae 2).
This information was included in the new version of the manuscript. Result Section, page 4, lines 107-108.
Reviewer
- Ivermectin is very potent at the nanomolar concentrations, however the authors used 5mg/ml of ivermectin (equal to 5.7 mM) they didn’t see any activity against egg hatch. The author needs to discuss such discrepancies of ivermectin activity.
Response: Dear Reviewer, we now have noticed that due to a mistake in Table 1 where two numbers were inverted related to Eggs and Larvae in Ivermectin. We think this mistake occasioned such confusion. Let me clarify this issue. The mean of eggs recovered in the group with Ivermectin was 81.41 and this means that the EHI was 100%. We hope this explanation will satisfy such confusion. Thanks
Reviewer
- Table 1, the authors reported after incubation of ~100 eggs with ivermectin the mean of recovered eggs is 0 and recovered larvae is 81 which mean that the % EHI should be 0. However, the table incorrectly indicate 100% for ivermectin %EHI. Please explain.
Response: We hope our previous response will help to clarify this issue.
Reviewer
- in the egg hatch inhibition test section in the material and methods the authors mentioned using thiabendazole at 0.1mg/ml as positive control (not ivermectin) however, no data was shown for thiabendazole.
Response: We are so sorry, you are absolutely right!
We wrote the word Thiabendazole by mistake. The correct drug was Ivermectin.
This error was corrected in the new version of the manuscript.
Reviewer
- the paper needs another round of English revision for better reading.
Response: Suggestion attended
Reviewer 4 Report
All corrections should be done
There is no recommendations
There is no ethical approval code or statement (should be )
Materials and methods should be written after introduction and before the results and discussion
LN/22---remove hyphen (HAE)
LN/36---add egg hatching inhibition , biological model to the keywords
LN/24---against L3---do you mean 3rd larval stage or what ????
LN/29---mention route of ivermectin injection
LN/39---add sheep as it is very common
LN/40---add breeding instead of production
LN/42&44----the author cited the references once time as a number and name in another time ---same style should be
LN/48---what about resistance against antithelmintic drugs---explain if there is any possibilities to bioaccumulation in muscle tissues ???
LN/51---Haemonchus contortus--- give more details about class ,genus ,species ,family ---etc
LN/52---cause several anaemias---enumerate types of anemia ???
LN/52-53---add emaciation and cachexia instead of in appetence and loss of body weight and remove hyphen between body and weight
LN/180-188---without references
LN/81-82----where is the figure ????????!!!!!!!!!
LN/97---table is very short
Results should be expanded
Discussion should be more concise
As volume , issue , number , and pages ----all available so no need for the link(s)---APPLY for ALL
Some cited references contained more than 6 authors (should be 6 at the maximum limit plus etal with the last ones ) for example(Ref(14,20,21,22,25----etc)
Some journal names were written abbreviated , while others were not (same style should be )
Author Response
All corrections should be done
There is no recommendations
Materials and methods should be written after introduction and before the results and discussion
Response: We tried to strictly follow the Instructions for authors of Pathogens. And we think we have adjusted to these recommendations in the structure of the manuscript.
Reviewer
LN/22---remove hyphen (HAE)
Response: This issue was attended.
Reviewer
LN/36---add egg hatching inhibition, biological model to the keywords
Response: These keywords were included in the manuscript. Abstract section, Page 1, Lines 35-36
Reviewer
LN/24---against L3---do you mean 3rd larval stage or what ????
Response: We think you are right. We have included L3 meaning into parenthesis. Abstract, page 1, line 25.
Reviewer
LN/29---mention route of ivermectin injection
Response: The study was only at in vitro level, no administration of Ivermectin was used.
Reviewer
LN/39---add sheep as it is very common
LN/40---add breeding instead of production
Response: Suggestions attended
Reviewer
LN/42&44----the author cited the references once time as a number and name in another time ---same style should be
Response: This was corrected.
Reviewer
LN/48---what about resistance against antithelmintic drugs---explain if there is any possibilities to bioaccumulation in muscle tissues ???
Response: The anthelmintic resistance of the commercial anthelmintic in livestock is evident around the world. There are reports on the bioaccumulation of some anthelmintics like ivermectin in muscle, liver, fat, milk and kidney. In our research we used Ivermectin only as a positive control in our studies under in vitro conditions.
Reviewer
LN/51---Haemonchus contortus--- give more details about class ,genus ,species ,family ---etc
Response: This information was included in the introduction section.
Reviewer
LN/52---cause several anaemias---enumerate types of anemia ???
Response: This sentence is incorrect. Haemonchus contortus cause severe anaemia in the small ruminants. This error was corrected in the new version of the manuscript.
Reviewer
LN/52-53---add emaciation and cachexia instead of in appetence and loss of body weight and remove hyphen between body and weight
Response: Suggestions attended.
Reviewer
LN/180-188---without references
Response: A reference that support the section “4.1. Plant material” was included in the manuscript.
Reviewer
LN/81-82----where is the figure ????????!!!!!!!!!
Response: Sorry!
The figure corresponding to the HPLC chromatograms was omitted by mistake. We have included this Figure in the new manuscript
Reviewer
LN/97---table is very short
Response: We don´t understand this comment.
The information displayed in the table 1 indicate the lethal concentrations 50 and 90 including the confidence intervals of the extract and fractions.
Reviewer
As volume, issue , number , and pages ----all available so no need for the link(s)---APPLY for ALL
Response: Suggestion attended
Reviewer
Some cited references contained more than 6 authors (should be 6 at the maximum limit plus et al with the last ones) for example (Ref(14,20,21,22,25----etc)
Response: Suggestion attended
Reviewer
Some journal names were written abbreviated, while others were not (same style should be )
Response: Suggestion attended
All references were reviewed and written according to the format of the journal.
Reviewer 5 Report
In vitro Anthelmintic Activity of a Hydroalcoholic Extract from Guazuma Ulmifolia Leaves Against Haemonchus Contortus
The work is ineresting and should be published. It should seek new products for the problem and brings us interesting results.
The methodology applied is classical and validated.
In Mexico, Rodríguez-Vivas et al. (2017) estimated the po- 42 tential economic losses caused by GIN, which reached 445 million dollars/per year [3]. 43.... remove this and make the problem global rather than local
Table 1. Results of Haemonchus contortus egg ... table titles should be improved and self-explanatory.
Conclusion of work: Remove what was written and put: These results suggest that G. ulmifolia leaves could be potential candidates for the control of H. contortus or other gastrointestinal parasitic nematodes. This is more summarized, real and according to the abstract.
Author Response
Reviewer 5
In vitro Anthelmintic Activity of a Hydroalcoholic Extract from Guazuma Ulmifolia Leaves Against Haemonchus Contortus
The work is interesting and should be published. It should seek new products for the problem and brings us interesting results.
The methodology applied is classical and validated.
In Mexico, Rodríguez-Vivas et al. (2017) estimated the potential economic losses caused by GIN, which reached 445 million dollars/per year [3]. 43.... remove this and make the problem global rather than local
Response: Suggestion attended.
Part of the introduction was rephrased
Reviewer
Table 1. Results of Haemonchus contortus egg ... table titles should be improved and self-explanatory.
Response: To better understand the title of table 1 was improved
Reviewer
Conclusion of work: Remove what was written and put: These results suggest that G. ulmifolia leaves could be potential candidates for the control of H. contortus or other gastrointestinal parasitic nematodes. This is more summarized, real and according to the abstract.
Response: Suggestion attended
This information was included in the manuscript.
Round 2
Reviewer 1 Report
Extract concentrations and fractions used in the EHI were high. In my opinion, you should have used more contractions, between 0-100% efficiency.
Normally use thiabendazole as positive control. In the first version of the manuscript you used thiabendazole. Why did you change to ivermectin?
Line 309 and 313: “On the other hand” are repeat in lines 309 and 313.
Line 371: in this paragraph there is no reference.
Line 446, 456: use italic font only scientific names.
Line 462: you can use third stage larvae (L3)
Line 527: how many walls and test repeats?
Line 530 – I think that “one” are use as “an”.
Author Response
Reviewer
Extract concentrations and fractions used in the EHI were high. In my opinion, you should have used more contractions, between 0-100% efficiency.
Response
Thanks for your kind suggestion. We think that the information displayed in table 2 could explain this comment. This table shows the effective concentrations required to inhibit the egg hatching. In future studies, we will test these effective concentrations to corroborate the ovicidal effect.
Reviewer
Normally use thiabendazole as positive control. In the first version of the manuscript you used thiabendazole. Why did you change to ivermectin?
Response
We are so sorry, you are absolutely right! We wrote the word Thiabendazole by mistake. The correct drug was Ivermectin.
Reviewer
Line 309 and 313: “On the other hand” are repeat in lines 309 and 313.
Response
The sentence “On the other hand” of the line 149 was removed
Reviewer
Line 371: in this paragraph there is no reference.
Response
One reference was added in this section (4.2. Hydroalcoholic extract and fraction obtaining)
Reviewer
Line 446, 456: use italic font only scientific names.
Response
Suggestion attended
Reviewer
Line 462: you can use third stage larvae (L3)
Response
Suggestion attended
Reviewer
Line 527: how many walls and test repeats?
Response
Each assay was tested by triplicate with four repetitions (n=4, four wells for treatment). This issue give us a total of twelve repetitions distributed into three experiments.
This information was included in the new version of the manuscript. This information has been inserted into the Materials and Methods section, page 9, lines 253-254.
Reviewer
Line 530 – I think that “one” are use as “an”.
Response
Suggestion attended
Reviewer 2 Report
Thank you for improving the manuscript as requested and for clarifying some critical points. I hope there will be future research to verify any toxicity and minimum effective dose to be tested later in an in vivo model.Author Response
Reviewer
Thank you for improving the manuscript as requested and for clarifying some critical points. I hope there will be future research to verify any toxicity and minimum effective dose to be tested later in an in vivo model.
Response
Thanks for your kind suggestion.